# Machine Learning Gene Signature to Metastatic ccRCC Based on ceRNA Network

**DOI:** 10.3390/ijms25084214

**Published:** 2024-04-11

**Authors:** Epitácio Farias, Patrick Terrematte, Beatriz Stransky

**Affiliations:** 1Bioinformatics Multidisciplinary Environment (BioME), Federal University of Rio Grande do Norte (UFRN), Natal 59078-400, Brazil; epitacio.farias.101@ufrn.edu.br (E.F.); beatriz.stransky@ufrn.br (B.S.); 2Metropolis Digital Institute (IMD), Federal University of Rio Grande do Norte (UFRN), Natal 59078-400, Brazil; 3Biomedical Engineering Department, Center of Technology, Federal University of Rio Grande do Norte (UFRN), Natal 59078-970, Brazil

**Keywords:** renal carcinoma, ceRNA network, transcriptional signature, machine learning, metastasis

## Abstract

Clear-cell renal-cell carcinoma (ccRCC) is a silent-development pathology with a high rate of metastasis in patients. The activity of coding genes in metastatic progression is well known. New studies evaluate the association with non-coding genes, such as competitive endogenous RNA (ceRNA). This study aims to build a ceRNA network and a gene signature for ccRCC associated with metastatic development and analyze their biological functions. Using data from The Cancer Genome Atlas (TCGA), we constructed the ceRNA network with differentially expressed genes, assembled nine preliminary gene signatures from eight feature selection techniques, and evaluated the classification metrics to choose a final signature. After that, we performed a genomic analysis, a risk analysis, and a functional annotation analysis. We present an 11-gene signature: *SNHG15*, *AF117829.1*, *hsa-miR-130a-3p*, *hsa-mir-381-3p*, *BTBD11*, *INSR*, *HECW2*, *RFLNB*, *PTTG1*, *HMMR*, and *RASD1*. It was possible to assess the generalization of the signature using an external dataset from the International Cancer Genome Consortium (ICGC-RECA), which showed an Area Under the Curve of 81.5%. The genomic analysis identified the signature participants on chromosomes with highly mutated regions. The *hsa-miR-130a-3p*, *AF117829.1*, *hsa-miR-381-3p*, and *PTTG1* were significantly related to the patient’s survival and metastatic development. Additionally, functional annotation resulted in relevant pathways for tumor development and cell cycle control, such as RNA polymerase II transcription regulation and cell control. The gene signature analysis within the ceRNA network, with literature evidence, suggests that the lncRNAs act as “sponges” upon the microRNAs (miRNAs). Therefore, this gene signature presents coding and non-coding genes and could act as potential biomarkers for a better understanding of ccRCC.

## 1. Introduction

Renal cancer is a group of neoplasms originating in the renal tissues and classified by cell type or histologic characteristics, such as clear-cell renal-cell carcinoma (ccRCC), papillary renal carcinoma (pRCC), and chromophobe renal carcinoma (chRCC) [1,2,3]. Due to the silent characteristic of this disease [4], the diagnosis at the metastatic state occurs in approximately 30% of ccRCC patients [5,6].

In a study of 537 ccRCC patients, The Cancer Genome Atlas (TCGA) consortium [7] characterized significant alterations in the ccRCC cohort. The changes include mutations in genes such as *VHL*, *PBRM1*, *SETD2*, and *BAP1*; the deletion of the q arm of chromosome 3; and distinct arrangements involving messenger RNA (mRNA) and microRNA (miRNA). These alterations signify crucial mechanisms in ccRCC. More recently, other studies revealed important roles for non-coding RNAs (ncRNAs), a class of RNAs that comprise approximately 80% of the transcriptome [8,9,10].

The functions of lncRNAs are determined by their interactions with DNA, proteins, or other RNAs and their cellular localization [9,10,11,12,13]. The lncRNAs can act as a (i) decoy or “sponge” modulating the effector of their targets, (ii) guide the enzyme modifiers of histones or chromatin, and (iii) respond to various stimuli [14,15]. In particular, the ligation of the lncRNA with the miRNA affects their targets, characterizing endogenous competition between the lncRNA and the mRNA target of the miRNA [9,10].

The proposed “Competing Endogenous RNA” (ceRNA) hypothesis was based on the idea of communication between miRNAs, mediated by the miRNA recognition elements (MREs), with mRNA, lncRNA, and other ncRNAs [16]. Alteration in the ceRNA networks is observed in cancer and other pathologies, associating them with biomarkers for metastasis and other clinical outcomes or therapeutic targets [10,17,18,19,20].

Research involving RNA expression generates extensive and intricate datasets. Integrating this data with clinical information through machine learning techniques could facilitate the extraction of patterns in gene expression, enriching our comprehension of gene functions within the biological context [21,22]. Among the vast applications of ML, methods of classification and prediction are commonly applied in health research [23,24]. However, the lack of feature selection associated with the outcome variable could influence the performances of the algorithms [25]. Feature selection involves the analysis of variables based on their impact on the outcome, eliminating irrelevant ones, and enhancing their consistency and relevance for the model [26].

Recently, studies investigated the ceRNA network and gene signature association in ccRCC. Most of them focus on the relationship between ceRNA, immune response regulation, and prognosis [27,28,29,30,31,32,33,34,35,36], and there is a lack of information about gene signatures involving the ceRNA network and metastasis in ccRCC.

In this study, we constructed a ceRNA network and generated a gene signature based on feature selection algorithms to classify the metastatic profiles of ccRCC patients. We achieved an Area Under the Curve (AUC) of 81.5% and an accuracy of 72% in the classification task. The signature was validated using an independent dataset, and the biological functions of its components were investigated in the ceRNA network. The flowchart shown in Figure 1 displays a summarized view of the discovery process for the novel Recursive Feature Elimination (RFE) gene signature of ccRCC.

## 2. Results

### 2.1. ceRNA Network

To construct the ceRNA network, we used the differentially expressed (DE) genes of the TCGA-KIRC (n = 602) project. This analysis resulted in 2842 mRNAs and 271 lncRNAs DE based on the thresholds of |log2FC| > 2 and *p*-value adjusted for the False Discovery Rate (FDR) < 0.01 (Appendix A). With these DE genes, we constructed the ceRNA network composed of 18 lncRNAs, 128 mRNAs, and 75 miRNAs (Figure 2). The miRNAs were included in the ceRNA network as described (Section 4.2).

The network structure explicitly reveals the connections between miRNAs, lncRNAs, and mRNAs. Within this inferred network, we have observed the presence of a diverse group of genes that share miRNAs and some lncRNAs. This characteristic points to a clustered organization, where these genes are related to a common outcome. Upon closer analysis of this cluster organization, we noticed a pattern in which one cluster is fully connected while others are sparser.

In order to evaluate the topology of our ceRNA network, we tested the fitness of the degree distribution to a power-law model, $P\left(k\right)=\;\propto \;k^{\alpha }$, resulting in α = 2.163. We performed a Kolmogorov–Smirnov test with our ceRNA network, and the distribution of our data did not fit strictly to a power-law model. Nevertheless, using the likelihood ratio test, our network fitted between a power-law and a log-normal positive distribution (Appendix A).

### 2.2. Feature Selection

With the expression data from the 221 ceRNA network genes and the metastatic classification from the 192 patients, we conducted balanced performance assessments. Subsequently, we executed a training process for feature selection and developed nine initial gene signatures (Figure 3). Among the feature selection techniques, only stepAIC did not converge. The Recursive Feature Elimination (RFE) shows an accuracy of 76.30% and a Kappa coefficient of 0.5663, showing a moderate level of agreement between the actual metastatic samples and the predicted ones.

We dismissed the outcomes from the stepAIC method and performed the first benchmark. The xgbTree presented the best result, with an accuracy of 80% during the training, 60% for the test, and 68.3% for validation. Employing the Youden statistics, we selected the top four signatures. The four signatures shared some genes, and we constructed the final signature through majority voting, composed of *INSR*, *PTTG1*, *BTBD11*, *RASD1*, *HECW2*, *HMMR*, *RFLNB*, *hsa-miR-130a-3p*, *hsa-miR-381-3p*, *SNHG15*, and *AF117829.1*. To evaluate gene importance in random forest, we present a multi-way importance plot (Appendix A).

We conducted the second benchmark to the constructed signature (Table 1), using the ICGC-RECA project as a test dataset. We observed an accuracy of 72%, an AUC (Appendix A) of 81.5%, and a Brier Score of 0.1955.

To highlight the separability of the data through the 11-signature genes, we applied k-means clustering to partition TCGA-KIRC samples into two groups (C1 and C2). Subsequently, for dimensionality reduction and visualization, we implemented principal component analysis (PCA). Predominantly, the metastatic samples (M1) are located within the positive range of the first dimension (C1), whereas the non-metastatic samples are positioned on the opposite side (C2) (Appendix A). The chi-squared test revealed a significant association (*p*-value = 0.007) between the metastatic samples and the cluster 1. The analysis of gene contribution with Cos2 to the sample characteristics shows that positively correlated genes are located in the first quadrant within the majority of metastatic samples (M1) and group C1 samples (Appendix A).

### 2.3. Integrative Analysis of the Transcriptional Signature Components

#### 2.3.1. Genomic Alteration Analysis

Performing a genome-level alteration analysis enables us to evaluate their impact on the gene product. The alterations can include changes in the genetic structure, disruptions in protein synthesis, or variations in the quantity of the gene product. We used the Maftools package to investigate single-nucleotide polymorphisms (SNPs) and copy number variations (CNVs) in the TCGA-KIRC cohort.

The missense mutation is the most frequent alteration in SNP data, with approximately 44 variants per sample and a prevalence of cytosine and thymine transversions. Moreover, ten samples showed mutations in signature-coding genes (Appendix A). Specifically, we detected missense mutations in the genes HECW2, BTBD11, INSR, and PTTG1; a frameshift deletion in BTBD11; and a multi-hit mutation in HECW2. However, the genes HMMR, RASD1, and RFLNB did not present any mutations.

The analysis of copy number variation reveals substantial and frequent alterations in chromosomes 1, 4, 5, 6, 7, 12, 17, 18, and 20 across the samples. Upon investigating the chromosome locations of our gene signature in the National Center of Biotechnology Information, we found that while our genes were situated in the chromosomes undergoing significant alterations, they were not specifically located in the regions exhibiting notable modifications (Appendix A).

#### 2.3.2. Risk Analysis

To evaluate the relationship between the expression level of signature genes, the metastatic development, and the survival status of the patients, we performed a risk analysis. Aalen’s additive regression shows a significant relationship between some genes from the gene signature and patient survival, such as (i) *AF1117829.1* (*p*-value = 0.0001627)*,* (ii) *hsa-miR-130a-3p* (*p*-value = 0.016), (iii) *hsa.miR.381.3p* (*p*-value = 0.027), and (iv) *PTTG1* (*p*-value = 0.020), see Appendix A.

The odds ratio analysis shows that the miRNA hsa-miR-130a-3p and the lncRNA AF117829.1 are the only ones that had significant associations, with *p*-value = 0.011 and *p*-value = 0.029, respectively (Figure 4).

#### 2.3.3. Functional Annotation Analysis

We performed a functional analysis using the signature-coding genes and the targets of the signature miRNAs against the Kyoto Encyclopedia of Genes and Genomes (KEGG) pathways.

When evaluating the targets from the miRNAs and their biological pathways, well-known oncology-related pathways, such as the PI3K-AKT signaling pathway, the p53 signaling pathway, the transforming growth factor-beta (TGF-beta) signaling pathway, renal cancer, and the HIF-alfa pathway, were also observed in a statistically significant manner (Appendix A).

The annotated biological processes were associated with cellular division regulation, such as chromatid sister separation, and chromosome segregation (Figure 5). The pathways annotated for miRNA targets were related to the cellular division process. Also, other pathways related to signal transduction, growth factors, and DNA polymerase I regulation were significantly enriched (*p*-value < 0.05) (Appendix A).

### 2.4. Gene Signature and ceRNA Network

As the signature construction was performed upon the genes from the ceRNA network, the evaluation of their location and their first neighbors could improve knowledge about their functions and the possible metastatic effects in the ccRCC.

The ceRNA network presents a cluster organization, and the signature genes are located in clusters with distinct properties. Two of the signature genes are located in cluster 1,while others resided in locations with dense interconnections, such as cluster 2. Additionally, certain areas contained only one gene, as observed in clusters 3, 4, 5, and 6. Table 2 presents the genes from the signature, their first ligands within the ceRNA network, and their cluster localization.

## 3. Discussion

In the present study, a transcriptional signature associated with metastatic development was constructed using feature selection techniques in conjunction with ceRNA network data for the cases of patients diagnosed with clear-cell renal-cell carcinoma. Additionally, the biological behavior of the genes comprising the signature was evaluated to understand their actions within the tumoral environment in ccRCC.

Regarding the network topology, our ceRNA presents a characteristic topology that does not follow the power-law degree distribution or represent a scale-free network. As indicated by Broido-Clauset [37] and Clauset, Shalizi, and Newman [38], the existence of scale-free topologies in the real world is rare, and most of them follow a log-normal distribution, like ours, or an exponential distribution once they have a heavy-tailed pattern.

### 3.1. Gene Signature

A set of nine feature selection methods produced preliminary gene signatures for the metastatic classification of ccRCC. The learning curves derived from RFE present a Kappa coefficient falling within the range of 0.41 to 0.6, which signifies a substantial agreement between the method’s outcomes and the data [39]. The combined application of these two metrics enhances classification accuracy [40].

Some benchmark models displayed overfitting, and we used the Youden statistics to select the models with the best sensitivity and specificity performances. Among the top four signatures, their Youden coefficients ranged from 0.13 to 0.18, approaching values closer to 1, signifying the optimal classification results [41].

The use of majority voting with the top four signatures results in the final signature of our work, composed of seven mRNAs: *PTTG1*, *BTBD11*, *HECW2*, *INSR*, *RFLNB*, *HMMR*, *RASD1*; two lncRNAs, *SNHG15* and *AF117829.1*; and two miRNAs, *hsa-miR-381-3p* and *hsa-miR-130a-3p*.

Validation with an external dataset is a process in the ML field used to evaluate model generalization [42]. We performed external validation to classify metastatic tissue using the gene expression of our signature. The training was performed with TCGA-KIRC and testing with ICGC-RECA, resulting in an accuracy of 72% and an AUC of 81.5%. Since the classification relied on health data, these evaluation metrics might not perfectly align with the objective of the study [43]. This limitation can be partially explained by the heterogeneity of the data, the sample size, and the inherent complexity of the biological process underlying the ccRCC. As far as we know, this is the first study that analyzes the relationship between a ceRNA network and a metastatic gene signature in ccRCC.

### 3.2. Validation and Biological Interpretation

#### 3.2.1. Genomic and Functional Alterations

The somatic alterations of the coding genes in the signature were more commonly associated with missense or frame_shift_del, except for HMMR and RFLNB. Regarding the copy number variations, the amplified or deleted regions were not in the exact location of the genes in the signature.

Analyzing the risk associated with survival or metastasis development showed a significant association between four genes in the gene signature. The lncRNA AF117829.1 and the miRNA hsa-miR-130a-3p were present in both analyses. The miRNA association is related to various cancers, such as bladder, breast, hepatocellular, glioma, and osteosarcoma [44,45,46,47,48,49]. Therefore, the presence of the PTTG1 and hsa.miR.130a.3p genes corroborate the literature, where in a situation of high expression, the prognosis is poor, and the hsa.miR.13p and hsa.miR.381.3p are associated with metastatic development. However, the lncRNA remains unknown, and these features could be added to its actions, which are still under study.

The functional annotation revealed diversified pathways. Both approaches, using the coding genes and miRNAs, highlighted biological processes associated with cell cycle regulation, controlling the separation and segregation of the sister chromatids, RNA polymerase II transcription, and the up-regulation and accommodation of the transcription activity of coding and non-coding genes [50], as well as processes related to cell–cell communication.

#### 3.2.2. Gene Cluster Analysis

The ceRNA network presents a cluster organization, showing dense regions with highly connected gene networks, others with more sparse networks, and some isolated small clusters. We perform a cluster-by-cluster analysis, using the signature genes and their first ligands (Table 2) to evaluate a possible role in metastasis development.

The first cluster is composed of two genes from the signature: the mRNA HMMR and the lncRNA AF117829.1, as well as the miRNA hsa.mir.361.5p and the mRNA POLE2. The Hyaluronan Mediated Motility Receptor (HMMR) is responsible for the regulation of tumor cell motility [51], and its knockdown reduced peritoneal metastasis in gastric cancer [52]. The role of lncRNA AF117829.1 remains unknown, but it was described as related to the proliferation, differentiation, and regulation of T-cell immunity [53,54], and its expression is implied with metastatic development and the worst prognosis of ccRCC patients. In this context, the lncRNA AF117829.1 could be acting as a sponge over the miRNA, impairing the degradation of POLE2 and HMMR, and promoting cell differentiation and metastasis development.

Cluster 2 presents the BTB Domain Containing 11 (BTBD11) gene from the signature. Its mechanism remains unknown, but it is described as a target in the TGF-beta pathway, responsible for cell cycle and apoptosis regulation [55]. The BTBD11 first ligands are the lncRNA MAGI2-AS3 and the miRNAs hsa.miR.374a.5p and hsa.miR.374b.5p isoforms. The lncRNA–miRNA interactions are related to tumor suppression in breast and hepatocellular cancers [56,57], suppressing proliferation, migration, and invasion. With the down-expression of the lncRNA, the miRNA can degrade BTBD11, negatively regulating the TGF-beta pathway and promoting tumor development.

The third cluster is located in the most dense region of the ceRNA network and presents five of the signature protein genes, as well as three lncRNAs and eight miRNAs directly linked to the former. The insulin receptor (INSR) regulates the insulin signaling pathway and activates the oncogenic PI3K/Akt/mTOR pathway, and its high expression is inversely associated with patient survival in ccRCC and gastric cancer [58]. Refilin b (RFLNB) is responsible for the epithelial–mesenchymal transition (EMT) and inhibits tumoral growth in neuroblastoma and pleural malignant mesothelioma [59,60,61]. HECT-Type E3 Ubiquitin Transferase HECW2 (HECW2) acts in apoptosis regulation, and its high expression is related to a good prognosis in ccRCC [62,63]. Ras-Related Dexamethasone-Induced 1 (RASD1) inhibits the RAS superfamily of the short GTPases and, in high expression, induces a decrease in cell growth, leading to apoptosis [64]. The increased expression of H19, C1RL-AS1, and AC005154.1 lncRNAs presented in the cluster suggests their roles as miRNA sponges (Table 2). The impairment or attenuation of miRNAs could potentially stabilize RFLNB and INSR expression, promoting tumor growth. Furthermore, the decreased expression of RASD1 implies that the miRNAs remain stable, also favoring a tumorigenic environment.

Cluster 4 is composed of the miRNA hsa.miR.381.3p and its targets. Coronin 1C (CORO1C) regulates apoptosis and cell cycle progression [65], acting as an oncogene in ccRCC and non-small-cell lung cancers [66,67]. The ATPase Family AAA Domain Containing 5 (ATAD5) is responsible for DNA duplication [68] and cell cycle regulation in neuroendocrine hepatic tumors [69]. The Arginine and Serine Rich Protein 1 (RSRP1) is involved in spliceosome assembly and has a good prognosis in breast cancer, but its biological mechanism is still unknown [70]. Ring Finger Protein 149 (RNF149) regulates ubiquitination and proteasomal degradation and is associated with pancreatic cancer [71,72]. The high expression of lncRNA AC016876.2 can promote the capture of the hsa.miR.381.3p, hence stabilizing the miRNA targets and facilitating tumor development.

Cluster 5 presents the signature Pituitary Tumor-Transforming Gene 1 (PTTG1), an oncogene that regulates sister chromatid separation [73]. The interaction with the miRNA hsa.miR.186.5 regulates the TGF-beta and MAPK pathways in breast cancer and ccRCC [74]. These pathways are associated with essential processes, such as tissue development, proliferation, senescence, migration, apoptosis, and cell differentiation [75,76]. The lncRNA AC021078.1 is involved in cell differentiation and DNA repair [77], and its high expression can negatively regulate the miRNAs, giving the PTTG1 the possibility to act in tumoral and metastatic progression.

The signature lncRNA Small Nucleolar RNA Host Gene 15 (SNHG15) is located on the sixth cluster and regulates the NF-kappa-B pathway that represses cell proliferation and the epithelial–mesenchymal transition (EMT) in ccRCC [78]. In cases of high expression, SNHG15 correlates to metastatic progression in colorectal and non-small-cell lung cancers [79,80]. This cluster is also composed of the protein coding genes’ NFKB Inhibitor Epsilon (NFKBIE) inhibitor of the NF-kappa-B signaling pathway associated with the inflammatory process in cancer [81,82], the Interleukin 2 Receptor Subunit Beta (IL2RB), and Cbp/P300 Interacting Transactivator With Glu/Asp Rich Carboxy-Terminal Domain 4 (CITED4), which are the regulators of the T-cell immune response and gene transcription, respectively [83,84], both, in cases of high expression, present a poor prognosis and are related to metastasis development. As the SNHG15 presented an elevated expression, it could indicate a sponge effect upon the miRNA on the cluster, promoting the normal activity of miRNAs-target.

Thus, the behavior observed in some genes from signature corroborates with the literature. SNHG15, hsa.miR.130a.3p, PTTG1, INSR and HMMR were described in a ccRCC environment, exhibiting the higher expression that induces metastasis and promotes cancer development. Conversely, lower expression of miRNA hsa.miR.381.3p is associated with a poor prognosis and linked to the development of metastasis. However, the remaining genes in the signature are reported in the literature across several other solid tumors, and play a crucial role in cancer and metastasis development.

## 4. Materials and Methods

### 4.1. Data

The RNA-seq and clinical datasets from the TCGA-KIRC project (n = 602) were downloaded from Genomic Data Commons (https://portal.gdc.cancer.gov/, accessed on 1 May 2023) [85] and UCSC Xenabrowser (https://xena.ucsc.edu/, accessed on 1 May 2023) [86] (University of California, Santa Cruz, CA, USA). For external validation, we used the dataset of ccRCC (n = 91 patients) from the International Cancer Genome Consortium (ICGC-RECA, accessed on 1 June 2023) [87] (Ontario Institute of Cancer Research in Toronto, Canada).

### 4.2. ceRNA Network Construction

The ceRNA network was constructed from the differentially expressed genes’ mRNAs and lncRNAs. We use the DESeq2 (v1.36.0) [88] package for the differential expression analysis between the normal (n = 72) and tumor tissues (n = 530) from the TCGA-KIRC cohort, with an absolute |log-fold change (LFC)| > 2 and an adjusted *p*-value (FDR) < 0.01.

With the differentially expressed genes, the ceRNA was constructed using the GDCRNATools package (v. 1.16.6) [89] associated with the starBase [90]. This database provides the iteration networks through numerous RBPs and RNAs, to supply the miRNAs shared by the differentially expressed lncRNAs and mRNAs from our KIRC dataset. The pair were selected using the following statistical analyses: (i) the hypergeometric test, where the probability of miRNAs shared by the lncRNA-mRNA pair was evaluated, observing the success of finding an association between the lncRNA–mRNA pair with the same miRNA; (ii) the Pearson correlation, used to measure the expression correlation between the lncRNA and the mRNA to understand the relation between them; and (iii) regulatory similarity, which will count the Pearson correlation and the total of miRNAs shared by the lncRNA–mRNA pair.

This analysis used a *p*-value threshold of 0.01 for the Pearson correlation and hypergeometric test and a value different from 0 for the regulatory similarity. The ceRNA network visualization was implemented using Cytoscape software (v 3.10.1) [91].

Once the network was constructed, its topology was evaluated following the Barabási-Oltvai [92] concepts of network biology, associating it with the likelihood ratio test based on the method of Broido-Clauset [37], using the package powerlaw [93] in Python (v 3.12).

### 4.3. Dataset Construction, Feature Selection, and Gene Signature Construction

The signature construction used the ceRNA network genes following the methodology of Terrematte and colleagues [94]. The gene signatures were produced using the feature selection techniques in Appendix A and the OmicSelector package (v1.0.0) [95].

Within the gene expression dataset from the TCGA-KIRC (n = 602), a missing metastatic classification in 30 patients was observed, causing their remotion, and due to the unbalanced characteristic of the metastasis classification of presence (M1) or absence (M0), a propensity matching score balance was performed, maintaining 190 patients, with 95 from each class.

This new dataset was split randomly into three new datasets, following the ratio of 60% for training (n = 114), 20% for testing (n = 38), and 20% for validation (n = 38). For the signature construction process, we used the following feature selection techniques: Recursive Feature Elimination (RFE) and two iterated versions, Boruta, the Generalized Linear Model (GLM), the Akaike Information Criterion (AIC), Linear Discriminant Analysis (LDA), Lasso, and ElasticNet.

To improve the construction of the signature and optimize computational efficiency, we performed hyperparameter adjustments to the feature selection. The RFE techniques used cross-validation with ten folds, using a window frame of 50 genes in each iteration, and the iterated RFE versions used a window frame of ten genes for the signature.

With the nine signatures constructed, a first benchmarking stage was performed to select the signature with the best metrics for metastatic classification using the test and validation datasets. The first benchmark compared the signatures using the following models: random forest (rf), the Generalized Linear Model (GLM), eXtreme Gradient Boosting (xgbTree), and the Support Vector Machine with a Radial Kernel (svmRadial), performed ten times to seek the best parameter adjustment for each of them. The metrics used to evaluate this benchmark were accuracy, specificity, sensitivity, and the Youden statistics.

To evaluate the signature generalization, the external dataset from the ICGC-RECA project (n = 91) was used with the mlr3verse package (v0.2.7) [96] to perform the second benchmark, applying the following classification techniques: random forest, naive Bayes, kNN, svmradial, and XGBoost. The evaluation metrics were accuracy, balanced accuracy, the Brier score, and the AUC. The validation process used the TCGA-KIRC for training and the ICGC-RECA for testing.

### 4.4. Somatic and Copy Number Alteration Analysis

The somatic alterations analysis was conducted using the Mutation Annotation Format (MAF) datafile, using the Maftools package (v2.16.0) [97], extracting information about (a) the types and classification of variations, (b) the variation quantity by sample, and (c) the top 10 genes altered.

The copy number variation analysis requires the construction of the GISTIC file. The Genomic Identification of Significant Targets in Cancer (GISTIC) pipeline [98] resulted in information about amplification and deletions within the data, analyzed via the Maftools package. To perform GISTIC analysis, it was necessary to obtain the segmentation file obtained from the GDC Data Portal [85] and the reference genome, version 41, from the GENCODE [99].

### 4.5. Risk Analysis

To evaluate the relationship between the expression level of the signature genes, the metastatic development, and the survival statuses of the patients, we performed a risk analysis. With the survival (v3.5.7) [100] and finalfit (v1.0.6) [101] packages, we executed Aalen’s additive regression and an odds ratio analysis, respectively. Aalen’s regression acts as a complementary, or alternative, form of the Cox model. In this method (Aalen Regression), the covariables associations and their effects are determined, taking into account the gene set association to the death event [102]. The odds ratio quantifies the strength of association between each of the analyzed covariables separately and the outcome (metastasis) [103].

### 4.6. Functional Annotation Analysis 

The identification of the pathways enriched by the genes of the signature was performed against the gene ontology [104], focusing on the biological processes and molecular functions, using the clusterProfiler package (v4.8.2) [105] and the mirPath platform (v3.0) [106] for the functional characterization of miRNAs from the signature. Enriched terms with *p*-values < 0.05 were considered statistically significant.

### 4.7. Development

This study was constructed using the R programming language (v4.2.0) with the RStudio platform (v4) hosted on the servers of the Multiuser Bioinformatics Center of the Metropolis Digital Institute at UFRN. The constructed codes are available in the GitHub repository (https://github.com/epfarias/transcriptonal_sig_ceRNA_KIRC, accessed on 27 July 2023).

## 5. Conclusions

This study aimed to build a transcriptional signature of clear-cell renal-cell carcinoma from differentially expressed genes that act as a competitor endogenous RNA network.

Using feature selection techniques for signature construction represents a promising application in this vast area of pattern recognition and machine learning. By integrating expression data with clinical information, we successfully constructed transcriptional signatures comprising multiple genes. The incorporation of evaluative metrics allowed us to gain valuable insights into the signature, assessing the metrics of the accuracy, sensitivity, and specificity of the signature in order to classify metastatic tissue expression. Using the external dataset permitted the examination of the signature generalization, thus validating its action as a metastatic classifier in clear-cell renal cancer.

With the cluster analysis, it was possible to know the actions performed by the signature genes within the cellular environment of clear-cell renal-cell carcinoma and how the effects of this regulatory process occur, indicating new roles for the lncRNA AF117829.1 and the mRNA RASD1. As research in the realm of lncRNA actions on cancer development undergoes constant evolution, our latest findings provide novel insights that illuminate promising avenues for future exploration. The dynamic nature of this field underscores the importance of our study, pointing toward potential directions for further investigation.

## Figures and Tables

**Figure 1 ijms-25-04214-f001:**
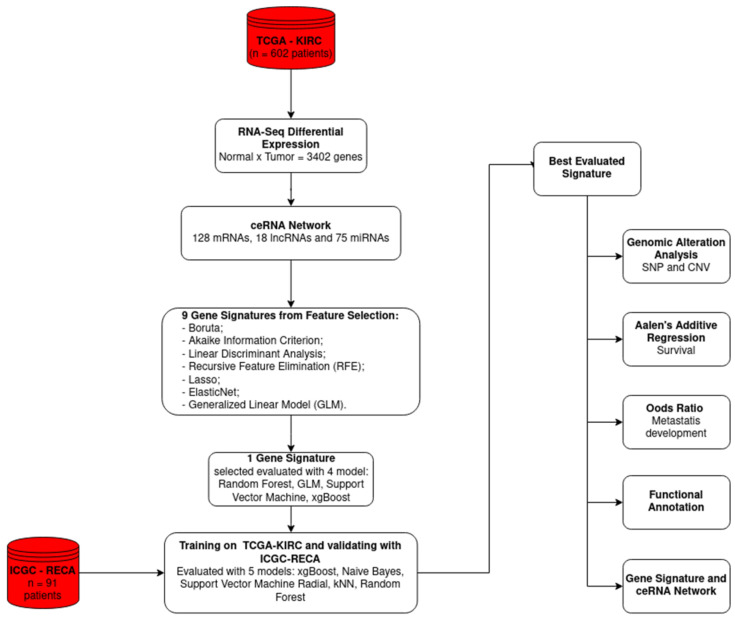
Flowchart of the current study to obtain a gene signature based on the Recursive Feature Elimination (RFE) approach. The datasets are indicated by the cylindrical shape; the white rectangles represent the steps of the study. TCGA-KIRC and ICGC-RECA are the ccRCC datasets.

**Figure 2 ijms-25-04214-f002:**
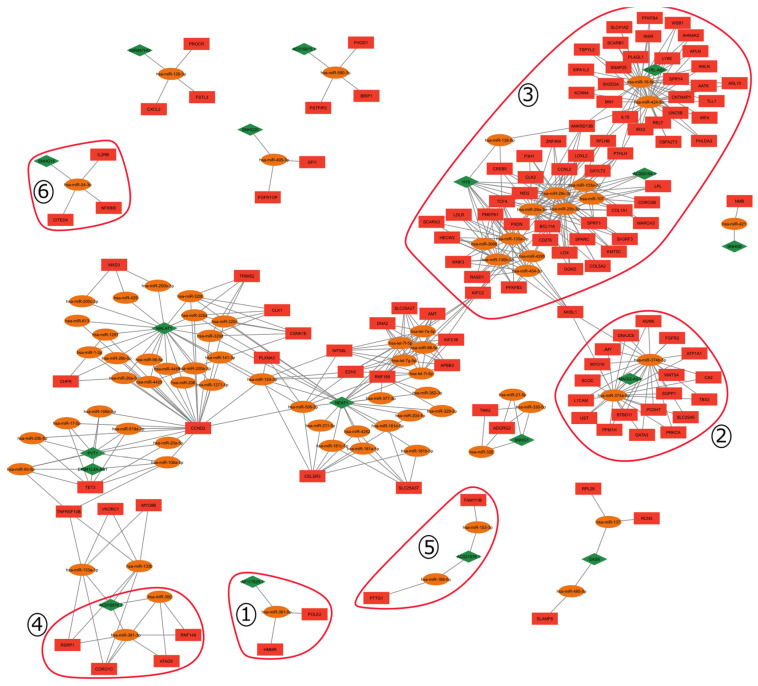
The ceRNA network was constructed based on the differentially expressed (DE) genes from ccRCC patients. The network is composed of 18 lncRNAs (green diamond), 75 miRNAs (orange ellipses), and 128 mRNAs (red rectangles). Individual clusters and clusters composed of gene signatures with their first neighbors are enumerated from 1 to 6, and highlighted by the red line circle.

**Figure 3 ijms-25-04214-f003:**
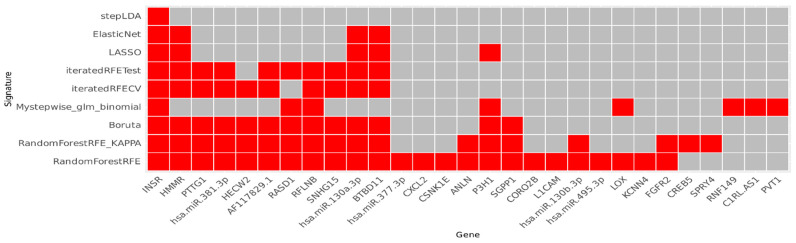
Heat plot with the 29 unique genes reported based on the nine preliminary gene signatures constructed. On the Y-axis are the models applied to the signature construction, and on the X-axis are the genes (red squares) from each obtained signature.

**Figure 4 ijms-25-04214-f004:**
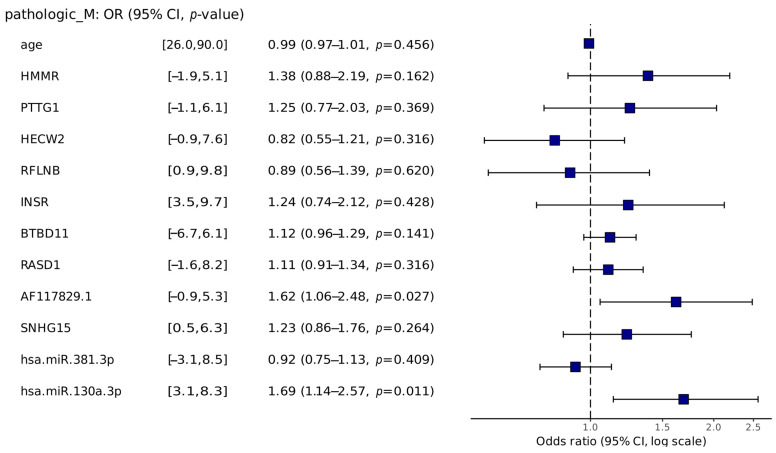
The odds ratio of each gene in the signature regarding metastatic development and the 95% confidence interval. The miRNA hsa-miR-130a-3p and the lncRNA AF117829.1 were the only ones that were significantly associated (*p*-value < 0.05).

**Figure 5 ijms-25-04214-f005:**
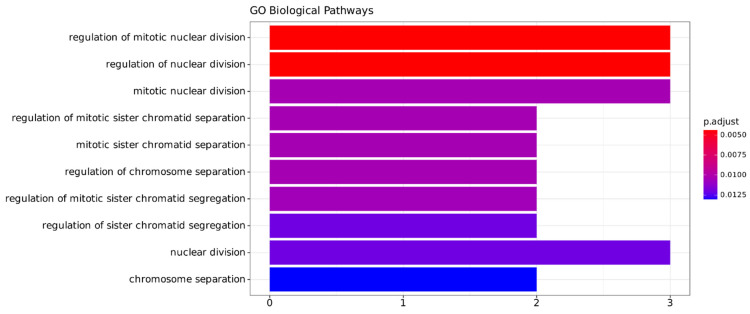
Functional annotation from the gene ontology, focusing on the biological process related to the seven coding genes from the signature (*HMMR*, *RASD1*, *RFLNB*, *PTTG1*, *INSR*, *HECW2*, *BTBD11*). On the Y-axis, the pathways annotated are listed, and the X-axis represents the gene count of the signature in each pathway.

**Table 1 ijms-25-04214-t001:** Metrics evaluated for validation using an external dataset.

Method	Accuracy	AUC	Brier Score
**Random forest ** ^1^	**72.2%**	**81.48%**	**0.1955442**
SVM	50%	66.67%	0.2500714
xgBoost	61.1%	62.34%	0.2343498
kNN	50%	61.72%	0.4817816
Naïve Bayes	50%	54.32%	0.5000000

^1^ Bold represents the best classification.

**Table 2 ijms-25-04214-t002:** Gene signature participants, their first ligands within the ceRNA network, and their cluster localization. Bold represents the gene from the signature’s first ligands.

Cluster	Gene	First Ligands
*1*	*AF117829.1*	*hsa-miR-361-5p, POLE2, **HMMR***
*2*	*BTBD11*	*hsa-miR-374a-5p, hsa-miR-374b-5p, MAGI2-AS3*
*3*	*HECW2*	** *hsa-miR-130a-3p* ** *, hsa-miR-130b-3p, hsa-miR-454-3p, hsa-miR-4295, hsa-miR-3666, H19*
1	*HMMR*	*hsa-miR-361-5p, POLE2, **AF117829.1***
*3*	*hsa-miR-130a-3p*	** *HECW2* ** *, WNK3, RASD1, PFKFB3, SCARA3, LDLR, PMEPA1, TCF4, PXDB, BCL11A, NHSL1, H19*
*4*	*hsa-miR-381-3p*	*RSRP1, CORO1C, ATAD5, RNF149, AC016876.2*
*3*	*INSR*	*hsa-miR-16-5p, hsa-miR-424-5p, C1RL-AS1.*
*5*	*PTTG1*	*hsa-miR-186-5p, AC021078.1*
*3*	*RFLNB*	*hsa-miR-29a-3p, hsa-miR-29b-3p, hsa-miR-29c-3p, hsa-miR-16-5p, hsa-miR-424-5p, H19, AC005154.1*
*3*	*RASD1*	** *hsa-miR-130a-3p* ** *, hsa-miR-130b-3p, hsa-miR-3666, hsa-miR-4295, hsa-miR-454-3p*
*6*	*SNHG15*	*hsa-miR-24-3p, IL2RB, NFKBIE, CITED4*

## Data Availability

This study utilized openly accessible datasets for analysis. The findings presented in this paper stem from information gathered by the TCGA Research Network. The TCGA-KIRC dataset (version 07-19-2019) can be accessed through the UCSC Xena Browser [86], while the ICGC-RECA dataset is available via the ICGC Data Portal [87].

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
