# Peer review of "Machine Learning Gene Signature to Metastatic ccRCC Based on ceRNA Network"

_ijms, 2024, doi:10.3390/ijms25084214_

Round 1
Reviewer 1 Report
Comments and Suggestions for Authors
I commend the author's dedication in preparing this manuscript. However, I have concerns regarding the novelty of this work, especially given the numerous publications that have already utilized the TCGA database for ceRNA, mRNA, and miRNA analyses. Additionally, relying solely on bioinformatics analysis without experimental validation may compromise the reliability of the findings.
Reviewer 2 Report
Comments and Suggestions for Authors
The manuscript by Epitácio Farias et al. builds a ceRNA network using TCGA-KIRC data to identify a gene signature for ccRCC associated with metastatic development and analyzes their biological functions. Using data from The Cancer Genome Atlas (TCGA), the authors assembled nine potential gene signatures from eight feature selection techniques and further presented an 11-gene signature. there remain some concerns that need to be addressed.
a) Major comments:
1. It is unclear the novelty of this study because the introduction lacks the discussion to compare with related works. Moreover, the results also didn't include any comparison with the identified signatures from other studies. For example, the previous work (https://www.mdpi.com/2072-6694/14/9/2111) for several authors presented 13-gene signature depending on the same dataset (TCGA-KIRC and ICGC-RECA). Is the 11-gene signature better than the 13-gene signature and the gene signatures from the other literature?
2. This study aims to identify the gene signature for ccRCC associated with metastatic development based on the ceRNA network. Thus, the ceRNA network construction is important, but there is no parameter tuning for building the network. For instance, why did the authors use a threshold of 0.1 for the Pearson correlation coefficient and hypergeometric test and 0 for the regulatory similarity? Generally, 0.1 is considered as having no or very low correlation, and a threshold of 0 for similarity also seems too loose.
3. Related to Point #2, most biological networks have been indicated to exhibit scale-freenet work characteristics [https://doi.org/10.1038/nrg1272]. Many previous works (e.g., https://doi.org/10.1186/1471-2105-9-559 and https://doi.org/10.1093/bib/bbad032) depended on measuring degree exponent (γ) and coefficient of determination (R2) values to determine whether the constructed network is close to real. The authors should evaluate the γ and R2 values of the current ceRNA network or rely on these values for parameter tuning.
4. Regarding functional annotation analysis (Figure 5a), only one count (i.e., one overlapping gene) between the query genes and genes in a specific pathway usually lacks statistical significance. The p-values should be adjusted by a multiple testing correction, such as the Benjamini–Hochberg correction.
5. It is confusing to use the same term to represent gene signature candidates (e.g., 9 gene signatures) and the gene signature (e.g., 11-gene signature). The authors should consider using different terms.
6. The top four signatures had low values of Youden's index, ranging from 0.13 to 0.18. The value of Youden's index (i.e., coefficient of Youden) ranges from -1 through 1 (inclusive). It has a zero value when a diagnostic test gives the same proportion of positive results for groups with and without the disease, i.e., the test is useless.
b) Minor comments:
1. In Figure 1, there are only red rectangles but no green rectangles.
2. The abbreviation of methods must be consistent in the main manuscript and figures, e.g., LAD vs. stepLDA in Figure 3; GLM vs. Mystepwise_glm_binomial.
3. In Figure 5a, does the gray dot mean non-significant? There is no corresponding legend.
4. In Figure S4a, there is no corresponding color legend for p-values.
Comments on the Quality of English LanguageSeveral abbreviations lack the definition. For example, WT and TP in Figure 1 and FDR.
Reviewer 3 Report
Comments and Suggestions for Authors
This study followed somewhat typical approaches to the construction of gene-signatures and subsequent analyses with the signature. Starting from differential expression analysis, the authors have constructed a ceRNA network, and they selected 9 genes that can be potential biomarkers for metastasis status from the genes in the network. Validating the signature with an independent dataset makes the finding more convincing. However, other analyses, or at least the presentation of them, are patchy, and there are a lot to be filled in about the details.
First, the 9 genes were claimed as potential biomarkers, but it is surprising that this manuscript did not discuss what is known about those genes in association with what were shown about those genes in this study.
Those 9 genes were selected by the majority voting. According to Figure 3, ‘P3H1’ was selected by 5 feature selection methods, while ‘HECW2’ was selected by 4. However, ‘HECW2’ was selected for the model, while ‘P3H1 was not. I wonder why?
The equation1 in page 4 may be a schematics of the 9-gene signature. But, it needs much more details. It could be assumed that the gene names mean their expression level. However, what is the coefficient for each gene, and how the estimates of the model are related to the metastasis class
In addition, the equation 1 in page 4 doesn’t contain the error term. I wonder if this is really true.
Also, Aalen’s additive regression included ‘age’ as a covariate, while the 9-gene signature model did not. Why?
It’s not clear how miRNAs were picked up as differentially expressed genes when RNA-seq data was used.
The risk analysis seems to be performed with one gene at a time, while the signature equation combined the behaviour of 9 genes together. Please clarify.
The second last sentence of the Abstract (“The gene signature analysis within … upon the miRNAs”) is a very strong statement. However, what is shown in this manuscript is the potential association of the 9 selected genes and the metastasis status. Whether the sponge action is at play in this association is also only a possibility given that those 9 genes were selected from the ceRNA network inferred from differentially expressed genes.
AU(RO)C curves were mentioned in multiple places, but they were not shown in the manuscript. Please add them with sufficient explanations.
Additional specific comments are as below:
In the last sentence of Abstract, ‘this gene signature identified new coding and non-coding genes’ isn’t a relevant statement. Identifying new genes is not what the gene signature in this study does.
Some abbreviations were used before their introduction (E.g., in the last paragraph of page 2, the abbreviation ‘RFE’ ).
In Figure 1 legend, the ‘green rectangles’ may actually mean ‘red rectangle’?
The FIgure 2 needs to be in higher resolution.
In Figure 3, there is only ‘stepLDA’, while ‘stepAIC’ was much more frequently used in the main text.
In the First paragraph of ‘2.3.1 Genomic Alteration Analysis’, ‘single-nucleotide variants (SNVs)’ is perhaps the proper term in the context of somatic mutations, rather than ‘single nucleotide polymorphisms’.
“odds ratio”, not “odd’s ratio”.
The section 4.4 implies that the GISTIC results were separately generated in this study. However, I believe GISTIC data should be already available from TCGA. Please clarify if they were indeed re-generated, and the detailed methods if that is the case.
Comments on the Quality of English Language
Many sentences are lengthy and hard to follow. Moderate-to-extensive level of editing is much recommended.
Reviewer 4 Report
Comments and Suggestions for Authors
Farias et al.'s research delves into the study of clear cell renal cell carcinoma (ccRCC), a cancer type known for its metastatic tendencies. While the influence of coding genes on metastasis is well-established, this study ventures into the role of non-coding genes, particularly competitive endogenous RNA (ceRNA). Leveraging data from The Cancer Genome Atlas (TCGA), the team constructs a ceRNA network and identifies an 11-gene signature associated with ccRCC and its metastatic development, hinting at their potential as biomarkers. The study's validity is reinforced by the use of an external dataset from the International Cancer Genome Consortium (ICGC-RECA). Genomic analysis pinpoints genes situated on mutated chromosome regions, some with significant impacts on patient survival and metastasis. Functional annotation analysis unveils pathways relevant to tumor development. Furthermore, the study posits an interaction between long non-coding RNAs (lncRNAs) and microRNAs (miRNAs), offering insights into the intricate mechanisms at play in ccRCC.
Nonetheless, several areas within the research necessitate improvement:
- igure 1's title references "green color," which does not correspond with the figure's content.
- The text omits information about the cohorts subjected to differential expression analysis.
- The meaning of "WT X TP" in Figure 1 remains unclear.
- The process of constructing the ceRNA network from differentially expressed genes lacks a comprehensive explanation.
- The legend for Figure 2 requires more scientific precision.
- Including ROC curves for all tested algorithms in the supplementary material is recommended for enhanced result clarity.
-
Furthermore:
- The term "Figure 3 Annoate Belongs" needs clarification to improve understanding.
- Figure 4's quality is compromised and requires enhancement.
Additionally, the integration of clinical information is praised for augmenting the clinical relevance of transcriptional signatures. However, providing more detailed information about the specific integrated clinical variables and their significance is recommended. Evaluative metrics should be disclosed to better understand the reliability and performance of the transcriptional signatures. The use of an external dataset for validation is commendable for ensuring research robustness. Lastly, the cluster analysis section lacks specific findings and the significance of discovered roles of lncRNA AF117829.1 and mRNA RASD1, necessitating further elucidation.
Using English editing tools or seeking assistance from a native English speaker can greatly enhance the quality of your English language.
Round 2
Reviewer 2 Report
Comments and Suggestions for Authors
Although the authors have made efforts to respond to my previous concerns, several remain that need to be addressed.
1. When I queried the keywords "Competing Endogenous RNA"+"Clear Cell Renal Cell Carcinoma"+"metastasis" in Google Scholar, there were 2,430 results as the related works. (https://scholar.google.com.tw/scholar?hl=en&as_sdt=0%2C5&q=%22Competing+Endogenous+RNA%22%2B%22Clear+Cell+Renal+Cell+Carcinoma%22%2B%22metastasis%22&btnG=). According to search results, several works have already studied the ceRNA action in ccRCC associated with metastasis. It is difficult to judge the novelty of this study if there is not enough discussion and comparison with the related works.
2. In the revised Figure S4a, there is still no corresponding color legend for p-values.
3. The abbreviation “FDR” was still not defined in advance but directly used.
Reviewer 3 Report
Comments and Suggestions for Authors
Thank you for the responses to my questions and comments.
However, much of them remain unclear to me, unfortunately.
My ‘comment 1’ was not about the association between those 11 genes with metastasis development. It was about comparing the new findings in this study with existing knowledge of those genes. It was done for a few of those genes, but only very briefly. It is very critical for this kind of study to discuss in detail on the selected genes (those 11 genes, in this case) regarding their known function and role in cancer development and how that compares to the findings presented in this study.
Regarding my ‘comment 3’, the equation still needs more explanation. This manuscript is about the machine learning model for classifying metastasis patients using existing data. So, the classification model needs to be described in a lot more detail. I.e., this is the area that needs to be substantially elaborated, along with the in-depth discussion as aforementioned.
First, the schematic representation of the signature needs better clarification. I can understand that the equation1 was used to train the prediction model in each of the methods in Table 1. But, details still have to be explained: What is ‘Class’? What does each gene symbol mean?
Once the 11 genes were selected, the authors should show how the data look with those 11 selected genes. PCA or MDS are often enough to give a sense of the data in regard to the classification, and normally work well in leading to the more refined classification methods.
Also, it is very strange to see that the ‘Accuracy’ and the ‘Balanced accuracy’ are identical for all 5 methods. I wonder if the authors had figured out why this happened?
Regarding the dataset, the DE analysis was performed between tumour and normal. But, the datasets were not clearly described in relation to metastasis.
Also, in relation to the DE genes, they were used to construct the ceRNA network, from which the 9 (or 11) genes were selected. However, those selected features were tested for the classification of metastasis. Wouldn’t there be genes in tumour samples that show differential expression between metastasis classes? If so, wouldn’t the set of such DE genes be a better starting point to identifying potential biomarkers for metastasis classes?
Regarding my ‘comment 6’, the inclusion of miRNAs in this analysis is still unclear in the Methods section. Also, the second sentence of the ‘2.1 ceRNA Network’ section implies that the analysis used 132 differentially expressed miRNAs. Please clarify.
Regarding my ‘comment 7’, I do not see any changes in section ‘4.5. Risk Analysis’, which is the relevant section in the Methods to the risk analysis.
Regarding my ‘comment 8’, I understand that the authors have made the assumption based on existing knowledge. But I suggested toning down that statement.
Regarding the response below:
“Response: Thank you for pointing this out. We agree that identifying new genes was not our principal aim, but we ended up finding two genes undescribed associated with metastasis development, like the lncRNA AF117829.1 and the mRNA RASD1. We point them out as new findings once they are undescribed in the literature.”
-> I think ‘identifying new genes’ means something different. It means finding a gene that was never known to exist. This study identified those 11 (known) genes ‘as’ potential biomarkers, and used them to construct a signature. The gene signature classifies samples into designated categories. So, my comment was that ‘signature’ and ‘identifying new genes’ do not match. This sentence should be revised with proper terminologies.
Comments on the Quality of English Language
Regarding my comment on the quality of English language, my concern still remains. Sentences do not read well in multiple places.
Round 3
Reviewer 3 Report
Comments and Suggestions for Authors
Thank you for the responses to my questions and comments.
I now have only a few comments to make this manuscript clearer.
The answer in ‘Response 2’ is simply what the ‘formula (1)’ says. My question was ‘which value goes into each gene symbol in the model?’
In relation to that, the 1st paragraph of page 4 says that the TCGA-KIRC samples were clustered into two groups by the 11-gene signature, and that one group may have something to do with metastasis. However, that statement was not followed up by any further details, such as the characteristics of samples in each group.
Also, the two groups are referred to as ‘C1’ and ‘C2’ only in Figure S4. Please use those group names in the main text, too.
The highlighted part of ‘2.4 Gene Signature and ceRNA network’ describes clusters 1-6, which are, I presume, the gene clusters in the ceRNA network shown in Figure 2. However, when the ceRNA network was first described in ‘2.1 ceRNA network’, there is no mention of the cluster numbering. So, it’s very hard to associate the individual clusters in section 2.4 with genes in Figure 2. Please make them clearer.
Section ‘3.2.1’ describes and discusses 6 gene clusters in the ceRNA network with the reported behaviour (up or down-regulation) of the signature genes in metastasis where applicable. However, it should be also added whether the reported behaviour of those genes in previous literature are consistently observed in this study.
“Odd’s ratio” still appears in multiple places. Please correct them to ‘odds ratio’.
